# The practice of reaction window in an electrocatalytic on-chip microcell

Hang Xia[1], Xiaoru Sang[1], Zhiwen Shu[2], Zude Shi[1], Zefen Li[3], Shasha Guo[4], Xiuyun An[1], Caitian Gao [5,6] ✉, Fucai Liu [3], Huigao Duan [2,6], Zheng Liu [4] ✉ & Yongmin He [1,6] ✉

To enhance the efficiency of catalysis, it is crucial to comprehend the behavior of individual nanowires/nanosheets. A developed on-chip microcell facilitates this study by creating a reaction window that exposes the catalyst region of interest. However, this technology's potential application is limited due to frequently-observed variations in data between different cells. In this study, we identify a conductance problem in the reaction windows of non-metallic catalysts as the cause of this issue. We investigate this problem using in-situ electronic/electrochemical measurements and atom-thin nanosheets as model catalysts. Our findings show that a full-open window, which exposes the entire catalyst channel, allows for efficient modulation of conductance, which is ten times higher than a half-open window. This often-overlooked factor has the potential to significantly improve the conductivity of non-metallic catalysts during the reaction process. After examining tens of cells, we develop a vertical microcell strategy to eliminate the conductance issue and enhance measurement reproducibility. Our study offers guidelines for conducting reliable microcell measurements on non-metallic single nanowire/nanosheet catalysts.

Given the energy and environmental crises caused by the extensive use of fossil fuels nowadays, pursuing clean and sustainable alternatives is highly desirable[1–3]. Electrocatalysis presents a promising green energy technology that converts available feedstocks, e.g., water, nitrogen, oxygen, and carbon dioxide, into high-value-added energy products like hydrogen, urea, hydrogen peroxide, and hydrocarbon[4–7]. To improve the catalytic efficiency, various strategies, such as strain[8,9], defect[10,11], doping[12,13], phase transition[14,15], and heterostructure[16,17], have been employed to develop efficient, low-cost, and stable electrocatalysts. However, when investigating catalytic properties and mechanisms, conventional testing methods in which materials spin- or drip-coated onto the glassy carbon electrode glassy carbon or conductive thin films still face limitations[18]. First, the measured results are based on the average data of thousands of catalysts with varied sizes, shapes, and structures (e.g., edge, basal, and defect sites), thus making it challenging to probe the catalytic property of a specific microscopic structure. Second, catalysts are commonly mixed with binders and conductive additives, forming complex interfaces inside the electrodes. Third, applying external fields, such as light, electric, and magnetic fields, to modulate the catalyst's properties is interesting yet challenging in conventional methods[19]. Therefore, exploring a promising technology could be critically important to precisely probe the electrocatalytic properties at a single-catalyst level.

[1]State Key Laboratory of Chemo/Biosensing and Chemometrics, College of Chemistry and Chemical Engineering, Hunan University, Changsha 410082, P. R. China. [2]College of Mechanical and Vehicle Engineering, National Engineering Research Centre for High Efficiency Grinding, Hunan University, Changsha 410082, P. R. China. [3]School of Optoelectronic Science and Engineering, University of Electronic Science and Technology of China, Chengdu 610054, P. R. China. [4]School of Materials Science and Engineering, Nanyang Technological University, Singapore 639798, Singapore. [5]School of Physics and Electronics, Hunan University, Changsha 410082, P. R. China. [6]Greater Bay Area Institute for Innovation, Hunan University, Guangzhou 511300, P. R. China. ✉e-mail: ctgao@hnu.edu.cn; z.liu@ntu.edu.sg; ymhe@hnu.edu.cn

Recently, thanks to advancements in micro/nano processing technologies and electronic devices, the on-chip microcell, as an electrocatalytic characterization technology, has attracted wide attention. Such a cell performed several fantastical functions[20]: (i) selective exposure of regions of interest. It enables distinguishing the activity among different zones and identifying the active sites at the micrometer scale[17]. For example, Liu and co-workers verified the activity of basal plane, edges, and boundaries of MoS$_2$ using the microcells during the hydrogen evolution reaction (HER) process[21]; (ii) external electric field tunning. The electric field could be precisely applied to catalysts using an on-chip configuration[22,23]. For example, Ding and co-workers reported the dynamic modulation of single-atom catalysis through electric fields, demonstrating a kind of electric field-induced charge polarization to optimize catalytic performances[24]; (iii) in situ monitoring the catalyst-electrolyte interface[25]. For instance, Duan and colleagues recently used electrical transport spectroscopy to reveal that smaller cations facilitate higher coverage and faster kinetics[26]. Although considerable progress has been made using on-chip microcells, it has been frequently observed that even if the same size of reaction windows were exposed on one catalyst (i.e., the same reaction surface), the microcells could give distinct performances, showing cell-to-cell variation in data.

Here, we identified a conductance problem raised from reaction windows to echo the current data variations observed between different cells. First, based on in situ electronic/electrochemical measurements and atom-thin nanosheets as model catalysts in microcells, we investigated their catalytic performance and corresponding electric properties through full-open and half-open reaction windows with the same sizes. We later demonstrated that a full-open window allows an efficient conductance modulation on non-metallic transition metal dichalcogenides (TMDs), which is an order of magnitude higher than a half-open window, thus promising better activity. In contrast, the metallic catalyst is not affected. We finally developed a graphene-layer-supported heterostructure strategy to eliminate such a conductance issue. This work clarified the reaction window-induced data variation in microcells, potentially providing guidelines for performing reliable microcell measurements on non-metallic catalysts.

## Results

### Reaction window in a microcell

Figure 1a illustrates the structure evolution of the on-chip microcell derived from the ion-gated transistor (IGT) in electronics. Based on this IGT structure, such a microcell was developed by replacing its ionic liquid with reactive electrolytes, such as H$_2$SO$_4$[27–29], HClO$_4$[25,30,31], phosphate buffer saline (PBS)[32], and KOH[26,33–35]. In a microcell, an insulating layer such as PMMA[36,37], photoresist[31,38], or SiO$_2$[39–41] films is necessarily used to make the reaction window, as shown in Fig. 1b. It mainly plays three roles: (i) prevent the leakage–current interface

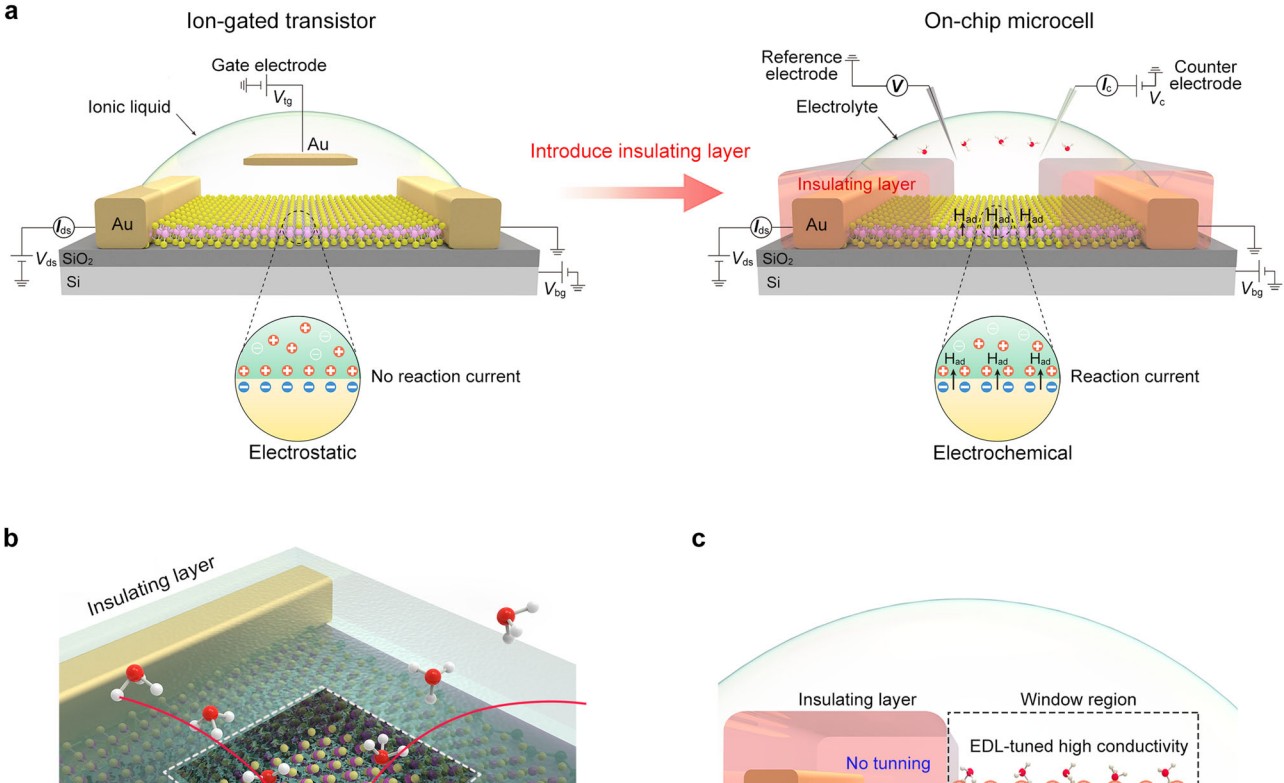

**Fig. 1 | Role of the reaction window in a microcell. a** Structure of an electro-catalytic microcell (right panel) derived from an ion-gated transistor (IGT) through introducing an insulating layer (left panel). Such a layer is used to make the reaction window that can prevent the leakage current from the metal electrodes as well as collect the reaction current during the electrocatalytic process. **b** Reaction window, playing three roles: (i) prevent the leakage-current interface; ii) make the reaction only happen at the regions of interest; and iii) measure the electrocatalytic surface area precisely. **c** Possible conductance issue due to the reaction window. Only the window region can be tuned to be highly conductive states by the electrical double layer (EDL), while the other regions (non-tunning) remain insulating or low-conductive, thus cutting off the conductivity channel of non-metallic catalysts and finally leading to poor performance.

raising from the metal electrodes[41]; (ii) make the reaction only happen at the regions of interest[42] to probe their catalytic activity; and (iii) measure the electrocatalytic surface area precisely. Benefiting this window, identifying active sites or even monitoring their change at a single nanosheet/nanowire is achievable, potentially guiding the design of high-efficiency catalysts[18–20].

However, it has also been frequently observed that even with the same-size reaction windows exposed, the microcells could give distinct performances on one catalyst, particularly for non-metallic catalysts. Here, we speculate that the microcell's conductance would account for such a cell-to-cell data variation, which is rooted in two points: (i) in situ conductance modulation. Non-metallic catalysts usually give a low conductivity but can be modulated to be highly conductive by an electric double layer (EDL) during the electrocatalytic measurement, which our previous self-gating works have verified[43]; (ii) the cut-off conductance channel due to the reaction window. Figure 1c shows that the channel only at the window region can be tuned to be highly conductive states by EDL, leaving other regions remaining low conductivity. As a result, the charge transport was still obstructed in the microcell's whole channel in electrocatalysis.

## Conductance issue caused by the reaction window

To elaborate on the above speculation, we first took monolayer $MoS_2$ as a typical catalyst example in our experiment (Supplementary Fig. 1), which has been widely used in hydrogen production[44–47]. To give a fair comparison, we fabricated a series of full-open and half-open microdevices with the same exposed area (see optical images in the middle panel of Fig. 2a), the former of which refers to the channels with full exposure to the electrolyte while the latter with partial exposure. Next, based on our four-electrode microcell configurations (see Supplementary Figs. 2 and 3), we conducted the in-situ electronic/electrochemical measurements to simultaneously collect its corresponding electronic and electrochemical signals. We also examined the morphological and structural stability throughout the device preparation process (see Methods, Supplementary Figs. 4 and 5) and fabricated the Pt microelectrode for pre-testing calibration (Supplementary Fig. 6).

For the full-open window (left panel in Fig. 2a), the conductance current dramatically increased with decreasing electrochemical potential, showing a threshold voltage at 0.2 V vs. reversible hydrogen electrode (RHE) and an on/off ratio of ~$10^5$ at 50 mV of drain-source voltage ($V_{ds}$) during HER. Those observations suggest a highly effective conductance modulation of monolayer $MoS_2$ (i.e., from insulating to conductive state), agreeing with the self-gating phenomenon in an $n$-type semiconductor (Supplementary Fig. 7). Importantly, it can achieve a high conductance, for example, 2.76 Ω mm at −0.10 V vs. RHE and 0.60 Ω mm at −0.55 V vs. RHE (see the detailed calculation methods in Supplementary Note 1), thus showing HER activity (onset potential of ≈ 0.29 V vs. RHE and Tafel slope of ≈165.05 mV dec$^{-1}$, see Fig. 2b) comparable with other works[14,38]. On the contrary, in the half-open one (right panel in Fig. 2a), the conductance modulation became much weaker, resulting in a much lower conductive state compared to full-open ones (see discussions about modulation effective in Supplementary Note 2). Correspondingly, it gives a poor HER performance (onset potential of 0.30 V vs. RHE and Tafel slope of ≈323.56 mV dec$^{-1}$), as shown in Fig. 2b. Similar phenomena of the full-open and half-open windows are also found in the multilayer $MoS_2$, as shown in Fig. 2c, d and Supplementary Figs. 8 and 9. Besides the HER process, we also observed similar phenomena in $p$-type $WSe_{1.8}Te_{0.2}$ nanosheet-based microcells with full-open and half-open windows during the oxygen evolution reaction (OER), see Fig. 2e, f and Supplementary Figs. 10 and 11.

On the other hand, we conducted in situ electronic/electrochemical measurements on metallic catalysts ($NbSe_2$, $PtTe_2$ nanosheets, and pure Pt film) with full-open and half-open windows. The characterizations of $NbSe_2$ and $PtTe_2$ nanosheets are presented in

Supplementary Figs. 12 and 13, and their transfer characteristics are shown in Supplementary Fig. 14a, b, respectively, both of which give a metallic behavior[48,49]. Figure 3a, c, e presents their corresponding electronic and electrochemical signals for full-open and half-open windows, respectively. Notably, we observed negligible conductance modulation in both full-open and half-open windows during the reaction process, possibly ascribing to their metallic behavior that is rarely affected by the EDL. Furthermore, their polarization curves in the full-open and half-open channel microcells closely resemble each other, demonstrating similar Tafel slopes and onset potentials, as shown in Fig. 3b, d, f, respectively.

Based on the above findings, we extracted the in situ conductance with the electrochemical potential (ratio of real-time conductivity to initial conductivity, $G/G_0$) for non-metallic and metallic catalysts, as shown in Fig. 4a, b, respectively. First, it is evident that, contrary to unchanged conductance in metallic ones, the $G/G_0$ of non-metallic catalysts increases with the electrochemical potential, suggesting an effective tuning of the Fermi level by EDL during electrocatalysis. Second, we found that $G/G_0$ (at −0.55 V vs. RHE) of non-metallic ones with full-open windows is near an order higher than that of the half-open window. Such a significant difference gives rise to distinct HER performances in the same materials with the same size windows (Fig. 2b, d). Those results indicate that the conductance problem raised from the reaction windows is non-ignorable or even critical for non-metallic catalysts.

To give insight into this problem, we analyzed the possible resistances along the charge transport pathway to clarify the window-induced difference in non-metallic microcells. As illustrated in Fig. 4c, in a full-open microcell, nearly the entire catalyst channel was under EDL-gating during the reaction, making all resistances tunable for achieving a highly conductive state (consisting of contact resistance, i.e., $R_{contact}$, and channel resistance, i.e., $R_{channel}$). Conversely, in the half-open microcell (Fig. 4d), only the exposed region can be tuned by EDL, leaving the other parts, e.g., the contact and the rest channel, remaining in their original insulating states. That is, the existence of non-tunable resistance in the half-open window (blue wavy line) fails to turn the whole microcell to a conductive state, thus resulting in poor performance. Taken together, we suggest that the position of reaction windows would cause non-tunable resistances that will result in a conductance issue in non-metallic catalysts, finally leading to cell-to-cell variation during measurements.

## Optimization of the electron transport pathway

Importantly, even though we have constructed full-open windows on the microcell each time in our experiment, a cell-to-cell variation was yet found after examining tens of the cells (see Supplementary Fig. 15). That is, not all full-open microcells can be tuned to be conductive for reliable testing. We attributed such a phenomenon to the fluctuation of the contact barrier (i.e., the charge barrier between the semiconductor channel and its above metal electrode), a common issue in semiconductor electronic fields[50,51]. Furthermore, owing to an insulating layer on its surface, this contact barrier cannot be well-tuned by EDL in the microcell compared to that in IGT (without an insulating layer covering). To abbreviate the influence of the contact barrier, we transferred the in-plane structure into a vertical one, shortening the electron-transport pathway for non-metallic catalysts in microcells. In our experiment, we prepared two kinds of heterostructure microcells, including $MoS_2$/Au and $MoS_2$/graphene, as shown in Fig. 5a, b, respectively. Their device fabrication processes were detailed in Methods and Supplementary Fig. 16. It should be noted that the $MoS_2$ catalysts in this vertical structure are still under EDL-gating during the reaction process, and the optimum thickness is expected to be lower than 20 nm; please see the discussions in Supplementary Note 3.

Figure 5c, d shows the optical images of our microcells with $MoS_2$/Au and $MoS_2$/graphene vertical heterostructures,

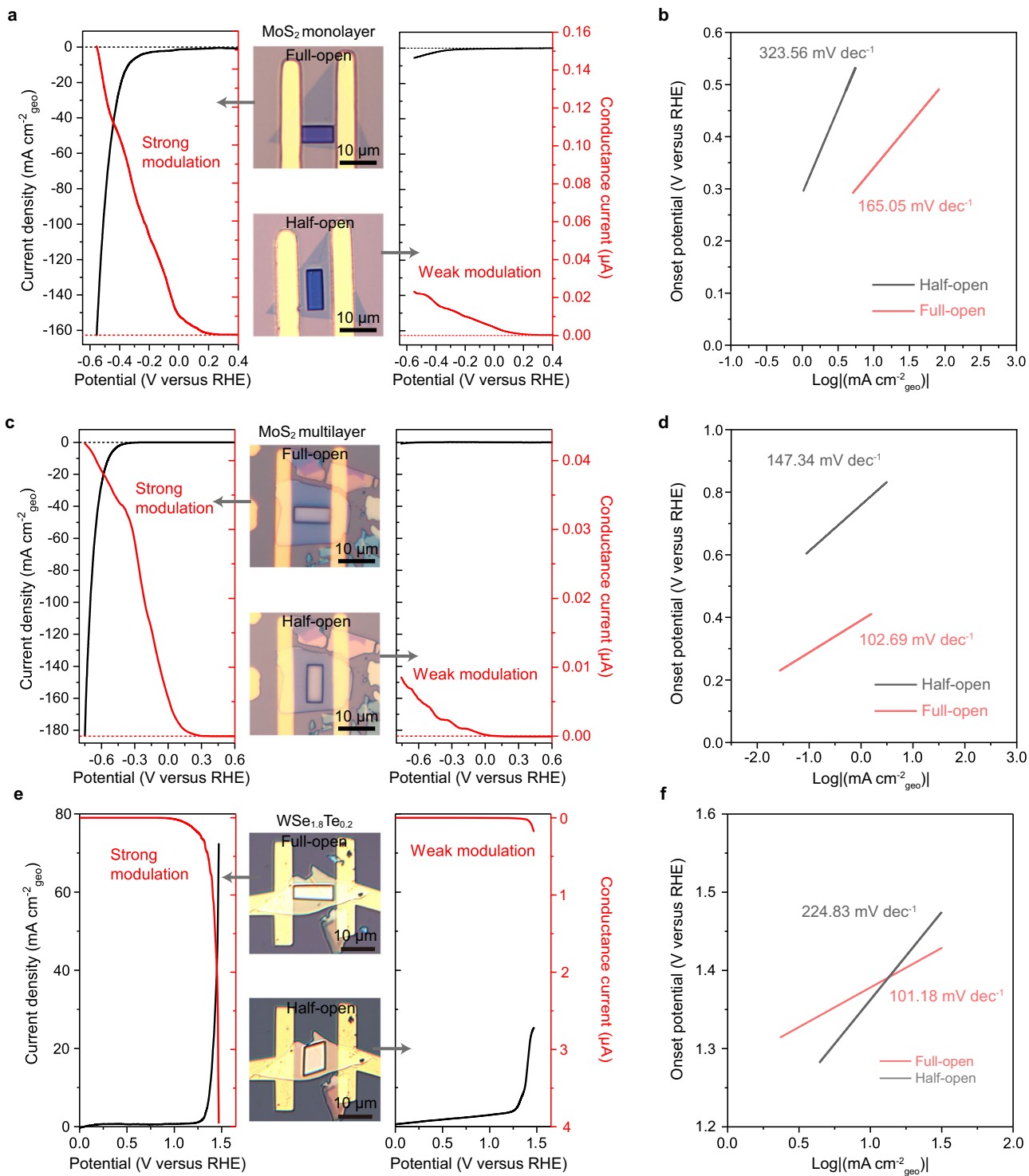

**Fig. 2 | In situ electronic/electrochemical measurement of non-metallic catalysts. a** The measurement on the non-metallic monolayer MoS$_2$ microcells with full-open and half-open windows. For full-open (left panel) and half-open (right panel) windows, the polarization curves (black line) and corresponding in situ conductance current curve (red line) are simultaneously collected. The optical images of full-open (top) and half-open (bottom) windows are shown in the middle panel. **b** The corresponding Tafel slopes of monolayer MoS$_2$ microcells with full-open and half-open windows. **c, d** The measurement on the non-metallic multilayer MoS$_2$ microcells with full-open and half-open windows (**c**) and their corresponding Tafel slopes (**d**). **e, f** The measurement on the non-metallic WSe$_{1.8}$Te$_{0.2}$ microcells with full-open and half-open windows (**e**) and their corresponding Tafel slopes (**f**). Note I: The bias voltages in our experiments were kept at 50 mV for monolayer and multilayer MoS$_2$ and 100 mV for WSe$_{1.8}$Te$_{0.2}$, to collect electronic signals and ensure minimal interference to the electrochemical signals. Note II: The window size is ~60 and ~50 μm² for MoS$_2$ and WSe$_{1.8}$Te$_{0.2}$ devices, respectively.

respectively. The in-plane cells with half-open and full-open windows were also used here as the references (Fig. 5e, f). Figure 5g shows their polarization curves during the HER process, and their corresponding Tafel slopes were also extracted in Fig. 5h. Those measurements provide several interesting observations. First, the MoS$_2$/Au vertical cell performs similarly to the in-plane cell with full-open windows, and both are superior to the in-plane one with half-open windows. Those findings indicate that the vertical cells

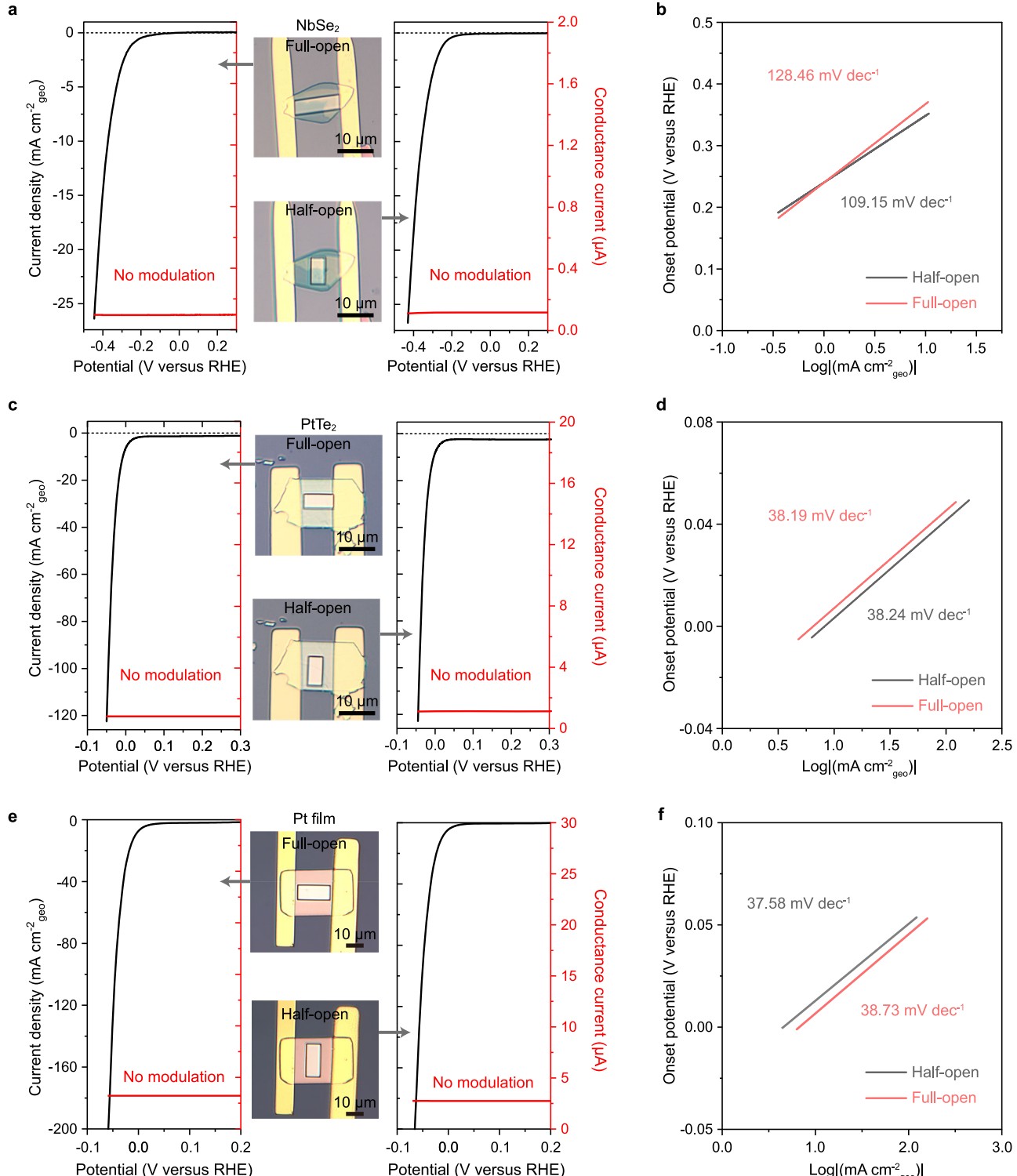

**Fig. 3 | In situ electronic/electrochemical measurement of metallic catalysts.**
**a**, **b** The measurement on the metallic NbSe₂ microcells with full-open and half-open windows (**a**) and their corresponding Tafel slopes (**b**). **c**, **d** The measurement on the metallic PtTe₂ microcells with full-open and half-open windows (**c**) and their corresponding Tafel slopes (**d**). **e**, **f** The measurement on the Pt film microcells with full-open and half-open windows (**e**) and their corresponding Tafel slopes (**f**). Note I: The bias voltages in our experiments were kept at 0.1 mV for NbSe₂, 1 mV for PtTe₂, and Pt film, to collect electronic signals and ensure minimal interference to the electrochemical signals. Note II: The window size is ~60 μm² for NbSe₂ and PtTe₂ devices and ~160 μm² for Pt film devices.

enable a similar charge efficiency to the highly-conductive in-plane one, eliminating the window-induced non-tunable resistance in the above in-plane structures. Second, compared to MoS₂/Au one, we observed improved MoS₂/graphene vertical cells' performances due to an enhanced charge-injection process in our graphene-based vertical structure. Notably, graphene layers have been widely applied in TMD-based electronic devices to avoid the Femi level pinning to the lower metal-semiconductor barrier, providing efficient electron injection for high-performance devices[52–56].

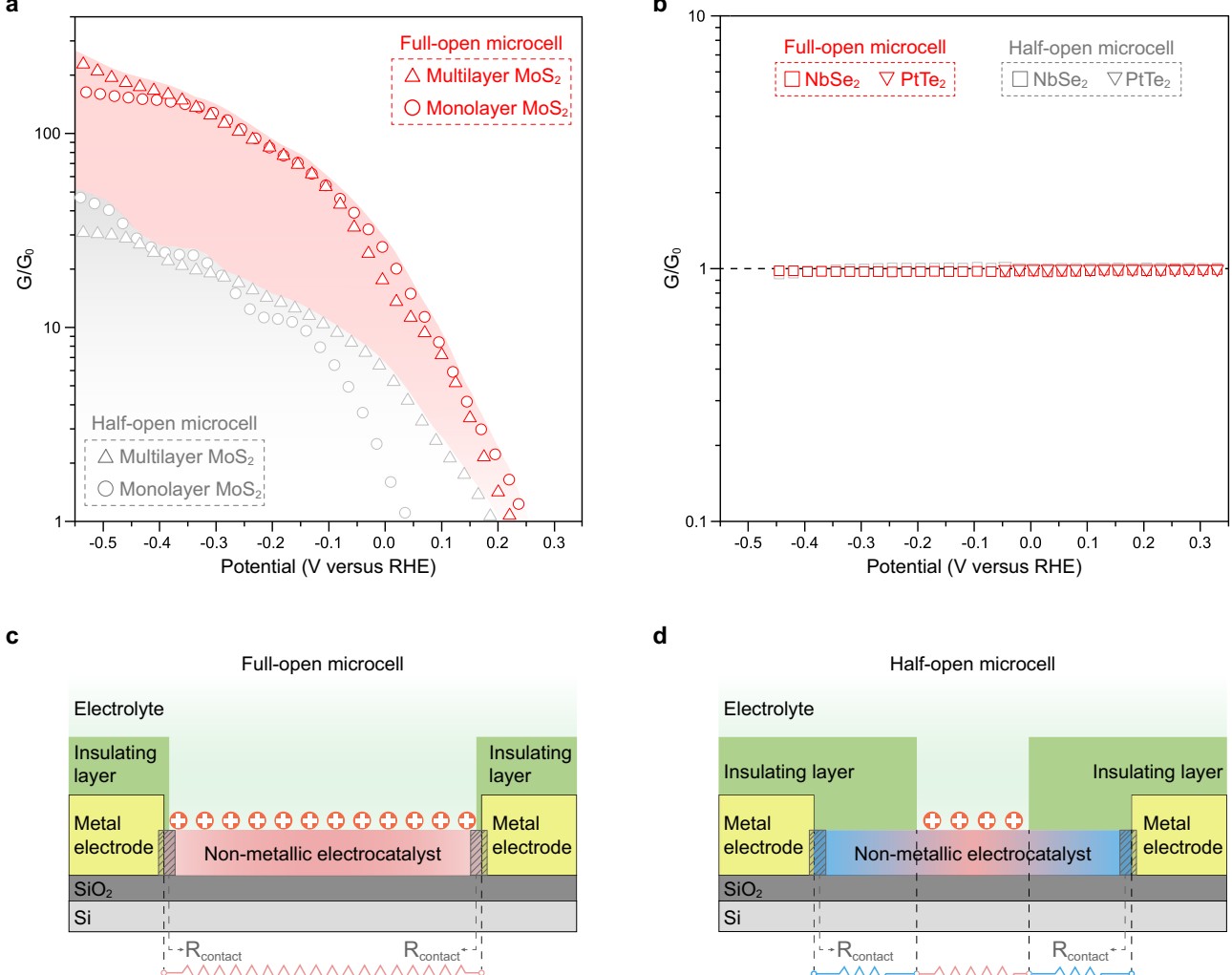

**Fig. 4 | Conductance issue caused by the reaction windows. a, b** In situ conductance ($G/G_0$) with the electrochemical potential for non-metallic (**a**) and metallic (**b**) microcells. The $G/G_0$ was defined by the ratio of real-time conductivity ($G$) to initial conductivity ($G_0$). The red and gray symbols represent the full-open and half-open microcells, respectively. **c, d** Schematic illustration of the possible resistances along the charge transport pathway in non-metallic microcells with half-open (**c**) and full-open (**d**) windows. $R_{contact}$, $R_{tunable}$, and $R_{non\text{-}tunable}$ represent the contact resistance, tunable resistance (under EDL modulation), and non-tunable resistance, respectively. Compared to full-open ones, the existence of non-tunable resistance (blue wavy line) in half-open windows fails to turn the whole microcell to a conductive state, thus resulting in poor performance.

## Reliability of optimized microcells

Furthermore, we have also carefully examined tens of in-plane and vertical microcells (Supplementary Fig. 17) and then extracted their corresponding onset potentials as well as Tafel slopes together in Fig. 5i. It is clear that vertical ones give remarkable reproducibility for the measurements in sharp contrast to unsatisfactory stability for in-plane ones. This is because the shortened pathway in the former can potentially exclude the window-induced negative influence on the catalyst channel (e.g., large contact or channel resistance) for non-metallic materials. Moreover, we observed that the graphene substrate-based vertical shows more reliable performances than that on the Au substrate, echoing the importance of charge injection discussed above. Notably, such a vertical-structure strategy can also maintain the structural stability of the MoS₂ catalyst during the cycling measurement (Supplementary Fig. 18).

Beyond hydrogen production, we also investigate the generality of this strategy to the OER. Our experiment fabricated the vertical microcells based on WSe₁.₈Te₀.₂/Au and WSe₁.₈Te₀.₂/graphene heterostructures (Supplementary Fig. 19a, b). Their polarization curves and corresponding Tafel slopes were shown in Supplementary Fig. 19e,

f, and the statistical results based on tens of microcells were also presented in Supplementary Fig. 19g. It can be seen that vertical ones not only give similar performance to the in-plane cell with full-open windows but also significantly enhance measurement reproducibility during the OER process. Together with the above similar phenomena in MoS₂ microcells in HER, our results demonstrate an effective graphene-based strategy for precise activity evaluation at a single catalyst and allow a fairer comparison among various catalytic sites.

## Discussion

We have revealed a conductance issue caused by the reaction windows in a microcell and developed a vertical strategy to solve the cell-to-cell data variation during measurements on non-metallic catalysts. Using in-situ electronic/electrochemical measurements, we clarified that the reaction window significantly affects the conductance of the catalyst channel. A full-open one enables a higher conductance modulation (nearly an order) than the half-open one, thus giving better activity during the HER process. In contrast, the metallic catalyst is not affected. Then, to eliminate such a conductance issue caused by windows, we developed a vertical cell strategy to shorten the charge pathway

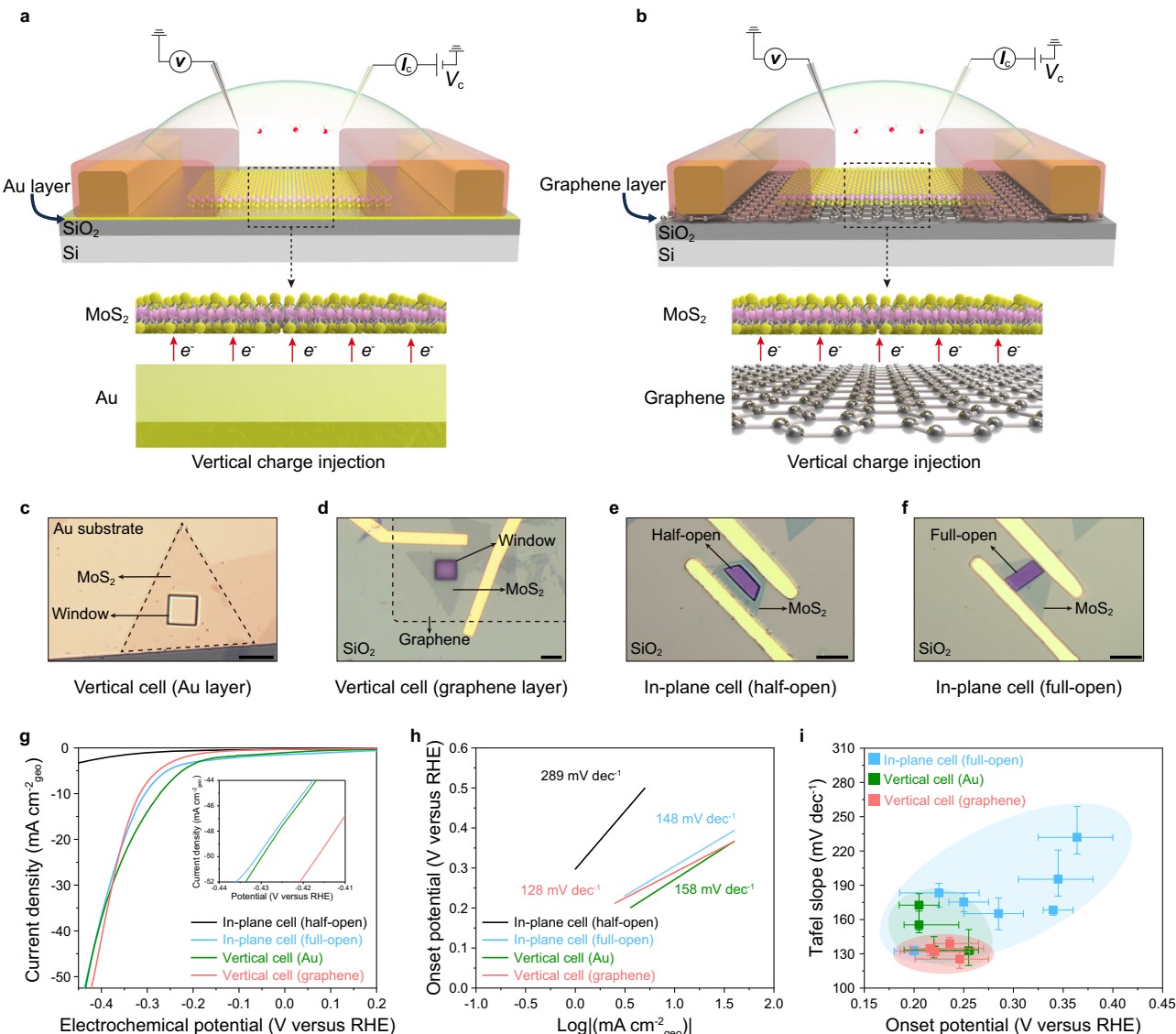

**Fig. 5 | Optimization of charge transport. a**, **b** Schematic illustration of the monolayer MoS$_2$ vertical microcells with the Au (**a**) and graphene (**b**) substrate layers. Insets in Fig. 5a, b show that those heterostructures potentially enable a vertical injection of electrons into the entire MoS$_2$ nanosheet to shorten the electron-transport pathway during the reaction process, thus eliminating the negative influence of the contact barrier in in-plane cells. **c**–**f** Optical images of various MoS$_2$ cells, including vertical cells with Au (**c**) and graphene (**d**) substrate layers, as well as in-plane cells with half-open(**e**) and full-open (**f**) windows. All scale bars are 10 μm. **g**, **h** Polarization curves of the current density (**g**) and the corresponding Tafel slopes (**h**) for those four kinds of MoS$_2$ cells. Inset in **g**: the zoomed-in view of the polarization curve. **i** Statistical measurement data of onset potentials and Tafel slopes obtained from tens of microcells for the HER process. The error bars represent the range of its values in our experiment.

and achieved remarkable reproducibility in measurements. Together, our work primarily aims at the conductance issue induced by the reaction window, providing critical guidelines for designing reliable microcells in electrocatalysis-related fields.

## Methods
### Fabrication of materials
**Monolayer MoS$_2$.** Monolayer MoS$_2$ was synthesized through a salt-assisted CVD method. A mixture of 0.2 mg NaCl and 5 mg MoO$_3$ powders loaded in a quartz boat was placed at the center of the quartz tube. The sulfur powder in an aluminum oxide boat was placed upstream. The growth temperature and time were 650 °C and 10 min, respectively. Ar, with a flow rate of 60 sccm was used as the carrier gas. The monolayer MoS$_2$ was then transferred to the SiO$_2$ (285 nm)/Si chip (pre-patterned 32 gold electrodes) through a PMMA-assisted transfer method for the next step of making the microcell.

**Exfoliation of TMD materials.** Multilayer MoS$_2$, NbSe$_2$, and PtTe$_2$ nanosheets were obtained by a poly-propylene-carbonate (PPC)-assisted mechanical exfoliation method[57]. In brief, the PPC anisole solution was first spin-coated on clean SiO$_2$/Si wafers (3000 rpm), followed by a baking process (80 °C for 2 min). Second, few-layer nanosheets were cleaved from their bulk crystals using blue tape (Nitto) and then transferred to those PPC-covered SiO$_2$/Si substrates by gently pressing them together for 30 s. Third, those nanosheet-loaded PPC films were peeled off from the SiO$_2$/Si substrate and then placed onto a homemade free-standing poly-dimethylsiloxane (PDMS) film. Fourth, those PPC films were transferred onto SiO$_2$ (285 nm)/Si chips pre-patterned with 32 gold electrodes and then removed by the acetone, leaving the few-layer nanosheets on the chips for the next step of making the microcell. In our experiment, the Ar plasma treated those mechanically exfoliated nanosheets for creating S, Se, or Te vacancies (2-4%) to provide active sites.

## Fabrication of microcells

**Fabrication of MoS$_2$, NbSe$_2$, and PtTe$_2$ nanosheet-based microcell.** Firstly, SiO$_2$ (285 nm)/Si chips with pre-patterned 32 gold electrodes were made through photolithography and electron beam evaporation. Secondly, those nanosheets were transferred to those chips, as introduced above. Thirdly, the laser direct writing method, followed by electron-beam evaporation, was used to make the gold electrodes (Cr/Au (5 nm/65 nm)) on the nanosheets. Finally, the device chip was spin-coated with PMMA film, and the reaction window of interest region was exposed by electron-beam lithography (EBL). Due to the PMMA's electrochemical inertness, the reactions only occurred on the exposed window.

**Fabrication of WSe$_{1.8}$Te$_{0.2}$ nanosheet-based microcell.** The whole preparation procedure is similar to (I). The only difference is that the adhesion layer of gold electrodes is Pd with high work function (Pd/Au (5 nm/65 nm)).

**Fabrication of 20 nm-thick Pt film-based microcell.** In order to ensure the accuracy and reliability of the micro-electrochemical measurement, we fabricated the Pt film device chips as the reference. The Pt/Ti (20 nm/2 nm) films were made on the pre-patterned SiO$_2$/Si by laser direct writing method followed by the electron-beam evaporation. The bottom Ti layer acted as the adhesion layer between Pt and SiO$_2$/Si substrate. The reaction window was carved on the PMMA by EBL for the HER measurements.

**Fabrication of monolayer MoS$_2$/gold film vertical heterostructure-based microcell.** First, the SiO$_2$/Si chip with pre-patterned 32 Au electrodes was made through photolithography and electron beam evaporation. Second, the monolayer MoS$_2$ grown by CVD was directly transferred to Au electrode patterns through the PMMA-assisted transfer method. Third, the reaction window was made on the basal plane of the monolayer MoS$_2$ nanosheet through an EBL process.

**Fabrication of monolayer MoS$_2$/graphene vertical heterostructure-based microcell.** First, high-quality and large-scale monolayer graphene was grown on Cu foils by CVD and then transferred to a pre-patterned SiO$_2$/Si substrate through the PMMA-assisted transfer method. Second, after spin-coated photoresist, the laser direct writing method and oxygen plasma were used to prepare uniform graphene patterns with the desired size, shape, and location. Third, monolayer MoS$_2$ nanosheets were transferred onto those graphene patterns using a mechanical alignment. Finally, the reaction window was carved on the basal plane of the monolayer MoS$_2$ nanosheet; see detailed procedures in Supplementary Fig. 16.

**Fabrication of WSe$_{1.8}$Te$_{0.2}$/graphene vertical heterostructure-based microcell.** First, a few graphene layers were mechanically exfoliated and transferred to the pre-patterned SiO$_2$/Si substrate. Second, a few layers of WSe$_{1.8}$Te$_{0.2}$ were exfoliated mechanically through the PPC-assisted methods and then transferred to the PDMS-glass slide stack. Third, the alignment between WSe$_{1.8}$Te$_{0.2}$ and graphene was realized via the optical microscopy and transfer stage. Thirdly, the gold electrode (Cr/Au (5 nm/65 nm)) was selectively deposited on the graphene. Finally, the reaction window was curved on the basal plane of WSe$_{1.8}$Te$_{0.2}$ through the EBL process.

## Micro-electrochemical measurement

A four-electrode configuration was used in our in-situ electronic/electrochemical measurement; see the equivalent circuit diagram in Supplementary Fig. 2. The high-purity graphite and the leakless Ag/AgCl micro-reference electrodes serve as the counter and reference electrodes, respectively. The other two electrodes connected to the nanosheets are used as the source and drain electrodes to simultaneously collect the conductance current (in-situ conductivity) and the electrochemical current during the HER process. Note: The conductance current is normally 10–1000 times the electrochemical current; In all experiments, only the exposed region of the nanosheets is attributed to the catalytic activity, and the left parts contacted with the electrolyte are isolated by the PMMA layer. The electrolyte is 0.5 M H$_2$SO$_4$, and the scan rate of electrochemical potential is 5 mV/point. The electrochemical current density is calculated by normalizing the current to the area of the reaction window on the nanosheet. In our experiment, the electrochemical reference voltage with respect to RHE was given by:

$$E_{RHE} = E_{Ag/AgCl} + 0.235 V \qquad (1)$$

## Data availability

The data that support the findings of this study are available from the corresponding author upon reasonable request.

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

## Acknowledgements

Y.H. acknowledges the National Key R&D Program of China (2021YFA1500900), the Fundamental Research Funds for Central Universities (531119200209), the National Natural Science Foundation of China (52203354 and 22272048), and the Guangdong Basic and Applied Basic Research Foundation (2023A1515012648). C.G. acknowledges the Guangdong Basic and Applied Basic Research Foundation (2023A1515012176).

## Author contributions

Y.H. conceived and initiated the project. Y.H., Zh.L., C.G., F.L., and H.D. supervised the project and led the collaboration efforts. Y.H. and H.X. designed the experiments. H.X. fabricated the electrochemical devices and performed micro-electrochemical measurements. X.S. grew the monolayer MoS2. Zh.S. carved the reaction windows through EBL. Y.H., H.X., and Zu.S. made the micro-electrochemical measurement setup. Ze.L. grew the single crystals. S.G. and X.A. did the Raman and AFM measurements. Y.H. and H.X. wrote the paper. All authors discussed the results and comments on the paper.

## Competing interests

The authors declare no competing interests.

 

## Additional information

**Supplementary information** The online version contains
supplementary material available at

Caitian Gao, Zheng Liu or Yongmin He.

**Peer review information** *Nature Communications* thanks Youwen Liu
and the other anonymous reviewer(s) for their contribution to the peer
review of this work. A peer review file is available.

