## [Peer Review File · Nature Communications]

The practice of reaction window in an electrocatalytic on-chip microcellREVIEWER COMMENTS

Reviewer #1 (Remarks to the Author):

In this manuscript, He and co-workers clarified the conductance problem in reaction window of non-metallic catalysts on the emerging on-chip microcell. The full-open window exposing the entire catalyst channel shows much more efficient modulation of conductance than the half-open window. Based on this finding, the authors developed a vertical microcell strategy to address the problem and enhance the measurement reproducibility. This work may guide the reliable microcell measurements of non-metallic single nanowire/nanosheet catalysts. The concept is interesting and may provide insight for future works. Some minor issues are listed below before publication.

1. Can the author quantitatively or semi-quantitatively rationalize the good conductance vs. bad conductance argument? Is there a threshold? Or the influence is subtler and more progressive? For example, in Figure 2c, the conductance modulation is relatively weak for half-opened windows, however the HER performance almost entirely disappeared, even after the channel is turned on at high potential. Is there a clear physical picture behind?
2. What are the Tafel plots in Figure S9?
3. Even with good conductivity and modulation, it seems the Tafel slopes vary a lot for all catalysts tested in this work? What could be the reason?
4. The analyses of Tafel slope are conducted at rather disparate zones. For instance, the Tafel slope of full-open CVD MoS₂ is obtained in the current density range of 6~60 mA cm⁻², while the Tafel slope of full-open multilayer MoS₂ is obtained in the current density range of 0.03~2 mA cm⁻². It is clearer to show the extended full Tafel plots to clarify the trend and mechanism.
5. When the effect of opening reaction window was studied on the same individual microdevice, the PMMA layer must be dissolved in organic solvent and recoated on the microdevice to open a new window. Since 2D nanosheets are extremely fragile, does the dissolution procedure in organic solvent affect the structure of the materials or the measurement? How reliable and repeatable is such measurement?

Reviewer #2 (Remarks to the Author):

He and co-authors identified a conductance problem in the reaction windows of non-metallic catalysts in the field of electrocatalytic on-chip microcell. They investigated this problem using in-situ electronic/electrochemical measurements and atom-thin nanosheets as model catalysts. The findings show that a full-open window, which exposes the entire catalyst channel, allows for efficient modulation of conductance. Furthermore, the developed a vertical microcell strategy to eliminate the conductance issue and enhance measurement reproducibility. This work can promote the further development of this field. Thus, I recommend this manuscript to be accepted after some revisions.

1. They co-authors show that a full-open window allows for efficient modulation of conductance. However, the author proposes to use a vertical microcell strategy to eliminate the conductance issue and enhance measurement reproducibility. Why can't we solve the repeatability problem directly with full-open window?
2. The developed a vertical microcell strategy to eliminate the conductance issue. For TMD electrocatalytic materials, the vertical conductivity of van der Waals layer is poor. How do the authors consider the effect of the interlayer conductivity of multilayer TMD materials on the conductance issue.
3. Figure 4i show that the vertical microcell strategy proposed by the authors can effectively increase device repeatability. However, for conventional electrocatalytic testing methods, the deviation of performance is still unacceptable. How does the author consider this issue?
4. The performance of monolayer MoS₂ obtained by device testing is worse than that of multi-layer MoS₂. Generally speaking, the performance of monolayer MoS₂ is considered to be better. How does the author consider this issue?
5. The Tafel slopes of MoS₂ obtained by on-chip microcell are all over 100 mV dec⁻¹ (, even as high as

300 mV dec⁻¹), which is much larger than conventional electrocatalytic testing methods. Why?
6. In order to ensure the integrity of the work, it is recommended that the author provide LSV curves.
7. Whether different window strategies (full-open, vertical microcell) affect the stability of catalyst.
8. Whether the author has tried the OER reaction and whether the strategy proposed by the author is suitable for the OER reaction.

Reviewer #3 (Remarks to the Author):

I have thoroughly read the manuscript entitled "The practice of reaction window in an electrocatalytic on-chip microcell". The manuscript shows that the authors compared the effects of full-open and half-open windows on the electronic and electrochemical performance of MoS₂ nanosheets. Then, a graphene-based vertical strategy is adopted to eliminate metal-semiconductor contact, which shortens the charge pathway and achieves remarkable reproducibility in measurements. However, there remain some problems in this manuscript. Thus, this manuscript is recommended to be published in Nature Communications after major revision. Please check the details in the following:

1. There may be a clerical error in the legend of Figure 1c. And there are some other grammatical and format mistakes in this manuscript. The authors should check it carefully.
2. The legend of Supplementary Figure 2a may be inaccurate. The tungsten needle is just like a wire connected to the working electrode.
3. "TMD" in the manuscript is not assigned a full name. Please provide the full name of TMD.
4. The authors mentioned that "Importantly, it can achieve a high conductance, for example, 2.76 Ω mm at -0.10 V (vs. RHE) and 0.60 Ω mm at -0.55 V (vs. RHE) ...". How are these resistor values calculated? It is better to write out the calculation process.
5. It is suggested to add the experiment of the full-open window and half-open window on Pt metal film as a comparison.
6. More related papers on single nanosheet electrochemical devices should be noticed. It is suggested that the authors cite the following relevant articles to make the manuscript more convincing.

Peiyao Wang et.al. Oxygen evolution reaction dynamics monitored by an individual nanosheet-based electronic circuit, *Nature Communications*, 2017, 8, 645.

Jiazhaohuang et.al. Back-Gated van der Waals Heterojunction Manipulates Local Charges toward Fine-Tuning Hydrogen Evolution. *Angew. Chem. Int. Ed.*, 2022, 61, e202203522.

Mengyu Yan et.al. Field-Effect Tuned Adsorption Dynamics of VSe₂ Nanosheets for Enhanced Hydrogen Evolution Reaction, *Nano Letters*, 2017, 17, 4109–4115.

Our Response to the Reviewers' Comments

Reviewer#1.....	2
Reviewer#2.....	12
Reviewer#3.....	22
References.....	25

Reviewer #1:

In this manuscript, He and co-workers clarified the conductance problem in reaction window of non-metallic catalysts on the emerging on-chip microcell. The full-open window exposing the entire catalyst channel shows much more efficient modulation of conductance than the half-open window. Based on this finding, the authors developed a vertical microcell strategy to address the problem and enhance the measurement reproducibility. This work may guide the reliable microcell measurements of non-metallic single nanowire/nanosheet catalysts. The concept is interesting and may provide insight for future works. Some minor issue are listed below before publication.

Response: We highly appreciate the reviewer's positive comments on our work, which help to improve the quality of our manuscript a lot.

Comment 1: Can the author quantitatively or semi-quantitatively rationalize the good conductance vs. bad conductance argument? Is there a threshold? Or the influence is subtler and more progressive? For example, in Figure 2c, the conductance modulation is relatively weak for half-opened windows, however the HER performance almost entirely disappeared, even after the channel is turned on at high potential. Is there a clear physical picture behind?

Response: Thanks for the reviewer's insightful suggestions about "*quantitatively or semi-quantitatively rationalize the good conductance vs. bad conductance argument? Is there a threshold?*". It is a good question, and we have been thinking about it when beginning the study of such an on-chip microcell field.

The electrocatalytic on-chip microcells have been developed from the top-gated field effect transistors (FETs) and electric double-layer transistors (EDLTs). Both of them contributed to this microcell with well-defined device structures and mature micro-/nano-processing technology, playing a vital role in its emergence and broad applications in electrocatalysis. As a result, the conductance-modulation-based working principle in on-chip microcells is also similar to FETs and EDLT, and a threshold should exist, as the reviewer suggested.

Here, we took MoS₂ catalysts in Figure 2a as an example, and extracted the subthreshold slopes (SS) of the full-open and half-open microcells, as shown in Figure R1. Note that, $SS = \frac{\partial V_g}{\partial(\log_{10} I_{ds})}$, where V_g is the electrochemical potential during the HER and I_{ds} is the conductance current¹. It can be seen that the full-open

one gives a SS of as low as 97 mV dec^{-1} , much better than that of the half-open one (215 mV dec^{-1}), verifying a higher efficiency of the conductance modulation in the former. It is worth mentioning that such an SS is also close to the ideal value in FETs (60 mV dec^{-1} , if the efficiency of electrostatic coupling between the gate and the channel region is 100% in the best case, then $\frac{C_s}{C_G} = 0$, and finally $SS = \ln(10) \frac{k_B T}{e} = 60 \text{ mV dec}^{-1}$). We added those discussions in Note 2 in Supplementary Information, and we also added the related description in the main text (Page 9 1st paragraph)

Figure R1. Transfer curves and subthreshold slope (SS) values of monolayer MoS₂ microcells with the full-open and half-open windows. The full-open window shows a much lower SS value than the half-open one.

Regarding the reviewer’s comment in Figure 2c “*the conductance modulation is relatively weak for half-opened windows, however the HER performance almost entirely disappeared, even after the channel is turned on at high potential.*”, we here showed an enlarged view of the measured HER curves in the half-open microcell, as shown in Figure R2. It can be seen that there is a slight signal of HER current, which is attributed to a low conductivity in the half-open microcell. We apologize for the misleading information and have added the zoom-in HER current data in Figure S8 in Supplementary Information.

Figure R2. The enlargement of HER performance of half-open microcell in Figure 2c. The left and right panels represent the origin and zoom-in HER polarization curves, respectively.

The on-chip electrocatalytic microcell, essentially, is based on the structure of the electronic device; as a result, good conductivity is a premise, especially for non-metallic catalysts. Recent work² has demonstrated that the EDL had a strong modulation on the conductivity of >30 types of non-metallic catalysts (self-gating phenomenon), covering a series of reactions such as HER, CO₂RR, OER, and oxygen reduction reaction (ORR). Moreover, such an EDL-modulation can also make non-metallic catalysts highly conductive or insulated, playing a critical factor in affecting the charge transport.

Based on the above distinct results in the full-open and half-open microcells as well as our previous works, *the possible physical picture* is the conductance channel cut-off by the half-open reaction window, as shown in Figure R3 (Figure 1c in the manuscript). That is, the channel only at the window region can be tuned to be highly conductive states by EDL, leaving other regions remaining low conductivity. As a result, the charge transport was still obstructed in the microcell's whole channel in electrocatalysis.

Figure R3. Possible conductance issue due to the reaction window. Only the window region can be tuned to be highly conductive states by the electrical double layer (EDL), while the other regions (non-tuning) remain

insulating or low-conductive, thus cutting off the conductivity channel of non-metallic catalysts and finally leading to poor performance. The left and right panels represent the origin and zoom-in HER polarization curves, respectively.

Comment 2. What are the Tafel plots in Figure S9?

Response: Following the reviewer's suggestion, we added the corresponding Tafel plots in Figure S15 in Supplementary information, also shown in the following (Figure R4).

Figure R4. Large cell-to-cell measurement variation in MoS₂ in-plane microcell with full-open windows.

Comment 3. Even with good conductivity and modulation, it seems the Tafel slopes vary a lot for all catalysts tested in this work? What could be the reason?

Response: We appreciate the suggestions from the reviewer. As known, the Tafel slope is a classical electrocatalytic parameter extracted from the corresponding LSV curve, involving the specific rate-limiting step of reaction kinetics. It serves as an indicator of electrocatalytic performance, and a smaller one is indicative of a better activity.

First, we made Pt microelectrodes and examined their corresponding Tafel slopes in hydrogen evolution reaction (HER), as shown in Supplementary Figure 6. It turns out a Tafel slope of 38 mV dec^{-1} , consistent with other reported works³. Those results ensure the reliability of our micro-cell measurement. In our work, we tested MoS_2 , NbSe_2 , and PtTe_2 nanosheets in the full-open windows, corresponding to the Tafel slopes of 165, 128, and 37 mV dec^{-1} .

Second, two factors are dominating the electrocatalytic performance. One is the *in-situ* conductivity of the catalysts during the reaction process. The other is the intrinsic catalytic activity, which affects the charge transfer kinetics at the electrocatalytic reaction interface. The latter can be described by Gibbs free energy of hydrogen adsorption (ΔG_{H^*}). In our work, although the full-open window could allow a high conductivity for all those catalysts during the reaction process, their intrinsic catalytic activities are yet distinct; please see ΔG_{H^*} of these catalysts collected from the published works⁴⁻⁶ in Figure R5.

Figure R5. ΔG_{H^*} of MoS_2 , NbSe_2 , and PtTe_2 catalysts in the volcano plot. Relationship of exchange current and Gibbs free energy of hydrogen adsorption (ΔG_{H^*}) via the density functional theory (DFT) calculation. Data obtained from references⁴⁻⁶. Note: These results are all based on the modeling of catalysts with vacancies in

non-metallic atoms, such as S, Se, and Te-vacancies.

Comment 4. The analyses of Tafel slope are conducted at rather disparate zones. For instance, the Tafel slope of full-open CVD MoS₂ is obtained in the current density range of 6~60 mA cm⁻², while the Tafel slope of full-open multilayer MoS₂ is obtained in the current density range of 0.03~2 mA cm⁻². It is clearer to show the extended full Tafel plots to clarify the trend and mechanism.

Response: Following the reviewer's suggestion, we show the full Tafel plots of full-open CVD MoS₂ (Figure R6a) and multilayer MoS₂ here (Figure R6b), and also added those data in Figure S9 in Supplementary Information.

Figure R6. Full Tafel plots of MoS₂ microcell in this work. a, The full Tafel plots of monolayer MoS₂ grown by CVD. **b,** The full Tafel plots of multilayer MoS₂ obtained by mechanical exfoliation.

Comment 5. When the effect of opening reaction window was studied on the same individual microdevice, the PMMA layer must be dissolved in organic solvent and recoated on the microdevice to open a new window. Since 2D nanosheets are extremely fragile, does the dissolution procedure in organic solvent affect the structure of the materials or the measurement? How reliable and repeatable is such measurement?

Response: We are very thankful for this careful suggestion. To investigate “*whether dissolution procedure in organic solvent affects the structure of the materials or the measurement?*”, we have chosen CVD-grown monolayer and mechanically-exfoliated multilayer MoS₂ as typical samples and characterized their possible morphological and structural properties through atomic force microscope (AFM) and Raman spectroscopy

during the device fabrication process. We added those data in Figures S4 and S5 in Supplementary Information and related discussions in the Experiment Section.

Figure R7 shows that the topography and Raman signals of 2D MoS₂ nanosheets did not change significantly during the multistep dissolution procedures. On the other hand, we noticed that excessive electron beam current or dose could cause irreversible damage to the 2D materials during electron beam lithography (EBL), as shown in Figure R8. In our experimental recipes, the electron beam current is 190 and 690 pA in Figures R7 and R8, respectively. Impressively, these findings remind us to protect 2D materials during the on-chip microcell fabrications with great care.

Figure R7. The morphological and structural characterization during the multistep dissolution procedures. a-d, The optical image, AFM image, and Raman signals of monolayer MoS₂ grown by CVD before and after the development. **e-p,** The optical image, AFM image, and Raman signals of multilayer MoS₂ nanosheets by mechanical exfoliation before and after the development. During the whole process, MoS₂ nanosheets' properties remained unchanged. Scale bar of optical images: 10 μm.

Figure R8. Irreversible damage of MoS₂ materials under excessive electron beam current or dose. a-c, the original optical and AFM image of MoS₂ nanosheets. **b-d,** the optical and AFM image of MoS₂ nanosheet after electron beam lithography (EBL) using excessive electron beam current or dose. There are visible small holes in MoS₂ material at the window regions. Scale bar of optical images: 10 μm.

Reviewer #2:

He and co-authors identified a conductance problem in the reaction windows of non-metallic catalysts in the field of electrocatalytic on-chip microcell. They investigated this problem using in-situ electronic/electrochemical measurements and atom-thin nanosheets as model catalysts. The findings show that a full-open window, which exposes the entire catalyst channel, allows for efficient modulation of conductance. Furthermore, the developed a vertical microcell strategy to eliminate the conductance issue and enhance measurement reproducibility. This work can promote the further development of this field. Thus, I recommend this manuscript to be accepted after some revisions.

Response: We highly appreciate the reviewer's positive comments on our work.

Comment 1. They co-authors show that a full-open window allows for efficient modulation of conductance. However, the author proposes to use a vertical microcell strategy to eliminate the conductance issue and enhance measurement reproducibility. Why can't we solve the repeatability problem directly with full-open window?

Response: This is an insightful question. In our preliminary experiment, we noted that even though we have constructed full-open windows on the microcell each time, a cell-to-cell variation was frequently found after examining tens of the cells in our experiment. As shown in Figure S15 in Supplementary Information, we have made ten microdevices with full-open windows, nearly half of which fail to be tuned highly conductive. *That is*, not all full-open microcells can be tuned to be conductive for reliable testing.

Along this line of thinking, we attributed such a phenomenon to the fluctuation of the contact barrier (*i.e.*, the charge barrier between the semiconductor channel and its above metal electrode), a common issue in semiconductor electronic fields^{7,8}. For example, due to fermi pinning, the contact between metal and MoS₂ usually delivers a high Schottky barrier.

More importantly, in an on-chip microcell, this contact barrier cannot be well-tuned by EDL in the microcell compared to that in IGT, because of an insulating layer covering the nanosheet channel. To abbreviate the influence of the contact barrier, we transferred the current in-plane structure into a vertical one, shortening the electron-transport pathway for non-metallic catalysts in microcells. Figure R9 shows statistical data of onset potentials and Tafel slopes obtained from tens of microcells based on vertical structures with Au and graphene

substrate layers and in-plane structures with full-open windows. Indeed, the vertical microcell gives a better reproducibility for the measurements, in sharp contrast to unsatisfactory stability for in-plane ones.

Figure R9. Statistical measurement data of onset potentials and Tafel slopes obtained from tens of microcells for the HER process. Blue, green, and red points represent in-plane cells with full-open windows, vertical cells with Au and graphene substrates, respectively.

Comment 2. The developed a vertical microcell strategy to eliminate the conductance issue. For TMD electrocatalytic materials, the vertical conductivity of van der Waals layer is poor. How do the authors consider the effect of the interlayer conductivity of multilayer TMD materials on the conductance issue.

Response: Thanks for this constructive suggestion. Usually, the resistivity of MoS₂ crystals along the c-axis is 3~4 orders of magnitude higher than that along the cleavage plane at 300 K⁹; please see Table R1 below. If there is no EDL-induced conductance modulation, undoubtedly, this will result in sluggish electron transport along the vertical direction during the reaction process.

Table R1. Reported resistivity of MoS₂ crystals

Along the cleavage plane (Ω mm)	Along the c-axis (Ω mm)
128	2.2×10^4

On the other hand, our previous work suggested that the surface conductance of catalysts can be significantly modulated under the electrical double layer (EDL)². Such a modulation would contribute to two behaviors:

i) high conductivity of the catalyst's surface. Our work shows that EDL enables tuning the conductance by nearly 6 orders of magnitude (Figure R10), and a highly-conductive state ($0.40 \Omega \text{ mm}$) can be obtained, much better than the intrinsic conductivity of TMD materials, e.g., vertical one ($2.2 \times 10^4 \Omega \text{ mm}$) or in-plane one ($128 \Omega \text{ mm}$).

ii) penetration depth beneath the semiconductor surface. In theory, such depth is estimated to be about tens of nanometers beneath the semiconductor surface (Figure R11a), according to the calculation formula given

$$\text{by}^2 |\psi_1(d)|^2 = \left| A \cdot \text{Ai} \left[\left(\frac{2m}{h^2 q^2 \epsilon^2} \right)^{\frac{1}{3}} \left(q \epsilon d - \left(\frac{h^2}{2m} \right)^{\frac{1}{3}} \left[\frac{9\pi q \epsilon}{8} \right]^{\frac{2}{3}} \right) \right] \right|^2, \text{ where } |\psi_1|^2 \text{ is the distribution of density of}$$

states, d is the depth beneath semiconductor surface, A is a proportionality constant that can be determined by normalization, m is the mass of electron, q is a unit charge, ϵ is the electric field strength. In experiment, we have examined such a penetration depth in the device with a bottom electrode configuration (Figure R11b), in line with the above theory results.

Those two behaviors would exclude the possible effect of the interlayer conductivity of multilayer TMD materials on the conductance issue. Furthermore, we strongly recommend that the thickness of 2D catalysts should preferably be less than 20 nm, according to our previous work. We add those discussions in Note 3 in Supplementary information.

Figure R10. The strong modulation ability of electric double layer (EDL). EDL enables tuning the conductance by nearly six orders of magnitude.

Figure R11. The penetration depth of surface conductance within the electrical double layer. a. Relationship between calculated carrier density and the penetration depth at the semiconductor-electrolyte interface. High carrier density prefers to form on the surface of the semiconductor electrode. The penetration depth is approximately 20 nm. **b.** HER current density of MoS₂ nanosheet decreases rapidly with the increasing thickness in vertical on-chip microcells. Inset: schematic illustration of penetration depth (top) and optical image of vertical MoS₂ on-chip microcell (bottom).

Comment 3. Figure 4i show that the vertical microcell strategy proposed by the authors can effectively increase device repeatability. However, for conventional electrocatalytic testing methods, the deviation of performance is still unacceptable. How does the author consider this issue?

Response: We greatly appreciate the reviewer’s insightful suggestions.

In the conventional electrocatalytic testing methods, catalysts are commonly mixed with binders (*e.g.*, Nafion) and conductive additives (*e.g.*, carbon black or graphene), forming complex interfaces. Moreover, the measured data are based on the statistical average of thousands of samples, failing to precisely evaluate the catalytic properties of a single unit (*e.g.*, single nanowire (NW)/nanosheet (NS)) and to distinguish the real active sites. Taking solution-exfoliated MoS₂ nanosheets as an example, the conventional testing methods (spin-coated or dropped on the conductive substrates) could give the statistical average of all active sites; it is challenging to distinguish which one (the basal plane, edges, or grain boundaries) have mainly contributed. Therefore, those complex interfaces and the statistical average data result in the deviation of performance under conventional electrocatalytic testing.

One potential way is to extract a single nanomaterial and precisely probe the catalytic behavior at a single-catalyst level. Along with advancements in micro-/nano-processing technology, an on-chip microcell has

recently emerged and has become a powerful tool in the electrocatalytic field. Such an on-chip microcell presents several fantastic features, including: i) identifying the active sites through reaction windows. Such a window is able to expose regions of interest on a catalyst using an insulation layer (such as PMMA); ii) regulation of reaction through applying an external electrical field. Based on a field-effect-transistor circuit, such a microcell also enables regulating the electrochemical reaction, e.g., activity or interface, by applying an electrical field using a back gate voltage; iii) *in-situ* monitoring electronic signals. Derived from a semiconductor electronic device, such a cell can in-situ probe the conductance of the catalyst during a reaction process; Finally, iv) design of catalyst interface. Using the micro-/nano-processing technology, various heterostructure interfaces, such as bottom-/top-electrodes, and interfinger electrodes, could be designed.

Comment 4. The performance of monolayer MoS₂ obtained by device testing is worse than that of multilayer MoS₂. Generally speaking, the performance of monolayer MoS₂ is considered to be better. How does the author consider this issue?

Response: This is a good question. Monolayer MoS₂ used in our work was grown by a CVD method, and this growth process naturally generates S-vacancies acting as the active sites for the later hydrogen production. On the other hand, multilayer MoS₂ is obtained by a mechanical exfoliation method in our experiment, whose perfect basal plane is usually HER-inert^{10,11}. For the HER measurement, we introduce some S-vacancies into the basal plane via the argon plasma to improve the electrocatalytic activity. This is the possible reason accounting for the above phenomenon. In the revised manuscript, we added detailed descriptions of the device fabrication in the Experiment Section.

Comment 5. The Tafel slopes of MoS₂ obtained by on-chip microcell are all over 100 mV dec⁻¹ (, even as high as 300 mV dec⁻¹), which is much larger than conventional electrocatalytic testing methods. Why?

Response: Thanks for the suggestion. In our work, the MoS₂ catalysts were made through CVD-growth or mechanical exfoliation, and S-vacancies serve as the active sites in HER. In full-open microcells (under good conductance modulation) or graphene-based vertical microcells, the Tafel slopes of the MoS₂ catalysts ranged

from 100 to 170 mV dec⁻¹; please see Figure R9. After reviewing other works, we note that those values are comparable to other MoS₂ with S-vacancies as the active sites, as shown in Table R2.

On the other hand, we noted that the microcells with Tafel slopes as high as 300 mV dec⁻¹ were actually the ones with half-open windows or some full-open ones (failing to be tuned). The observed poor performances would attributed to their low conductivity.

Table R2. Tafel slopes of the S-vacancy-based MoS₂ catalysts collected from other reports.

Synthesis method	Testing substrate	Tafel slope (mV dec ⁻¹)	Active sites	Reference
Ultrasonic exfoliation	Glassy carbon electrode	130	S-vacancies	12
CVD	Glassy carbon electrode	342	S-vacancies	13
Li-intercalated exfoliation	Glassy carbon electrode	91	S-vacancies	14
Ultrasonic exfoliation	Glassy carbon electrode	199	S-vacancies	15
CVD	Glassy carbon electrode	118	S-vacancies	16
Hydrothermal method	Glassy carbon electrode	93	S-vacancies	17
Ultrasonic exfoliation	Glassy carbon electrode	133	S-vacancies	18
Hydrothermal method	Glassy carbon electrode	135	S-vacancies	19
CVD	Not mentioned	98	S-vacancies	11
Solvothermal method	Glassy carbon electrode	157	S-vacancies	20
CVD	Gold substrate	136	S-vacancies	21
Not mentioned	Glassy carbon electrode	170	S-vacancies	22

Comment 6. In order to ensure the integrity of the work, it is recommended that the author provide LSV curves.

Response: Thanks for the reviewer's careful suggestion. We have provided the LSV curves corresponding to Figure 5i, as shown in Figure R12. We also add those data in Figure S17 in Supplementary Information.

Figure R12. Typical polarization curves of three kinds of monolayer MoS₂ on-chip microcells. Blue, green, and red curves represent the HER performance of in-plane cells with full-open windows, vertical cells with Au substrate, and graphene substrate, respectively.

Comment 7. Whether different window strategies (full-open, vertical microcell) affect the stability of catalyst.

Response: We appreciate the reviewer's suggestion. We examined the stability of catalysts in full-open in-plane microcells and vertical ones, as shown in Figure R13. It can be seen that the monolayer MoS₂ presented a stable HER performance with no obvious degradation in in-plane and vertical microcells after long-cycling CV measurement. We add those data in Figure S18 in Supplementary Information.

Figure R13. Stability of monolayer MoS₂ in full-open and vertical microcells. **a** and **c**, The comparison of optical images before and after 50 times cyclic voltammetry in the full-open (**a**) and vertical (**c**) microcell. **b** and **d**, The stable HER performance of in-plane (**b**) and vertical (**d**) microcell during long-cycling CV measurement. Inset in (**b** and **d**): the zoomed-in view of the corresponding polarization curve. Scale bar: 10 μm .

Comment 8. Whether the author has tried the OER reaction and whether the strategy proposed by the author is suitable for the OER reaction.

Response: This is a very constructive suggestion, improving the generality of our window strategy. Here, we took the $\text{WSe}_{1.8}\text{Te}_{0.2}$ nanosheet used as an OER TMD catalyst as a typical example². Figure R14 presented the Raman spectrum of $\text{WSe}_{1.8}\text{Te}_{0.2}$ used in our experiment. The back-gate measurement verified the *p*-type transfer characteristic (Figure R15), which would be suitable for OER, as our previous self-gating phenomenon suggested.

With the above characteristics in mind, we first conducted *in-situ* electronic/electrochemical measurements on the full-open and half-open microcells, as shown in Figure R16. It was observed that the $\text{WSe}_{1.8}\text{Te}_{0.2}$ microcell with full-open window shows a higher effective conductance modulation than that with a half-open one. As a result, it could achieve a high conductive state, and thus shows a better OER activity (72 mA cm^{-2} at 1.47 V vs. RHE and Tafel slope 101 mV dec^{-1}), compared to the half-open microcell (25 mA cm^{-2} at 1.47 V vs. RHE and Tafel slope 224 mV dec^{-1}) with low conductance. Those results are similar to non-metallic MoS_2 catalysts measured in our work, again verifying the conductance problem caused by the reaction window in the on-chip microcell.

Furthermore, to avoid the influence of the contact barrier, we used the vertical charge transport strategy on the $\text{WSe}_{1.8}\text{Te}_{0.2}$ for OER. We prepared two kinds of vertical microcells, including $\text{WSe}_{1.8}\text{Te}_{0.2}/\text{Au}$ and $\text{WSe}_{1.8}\text{Te}_{0.2}/\text{graphene}$, as shown in Figures R17a and R17b, respectively. The in-plane half- and full-open microcells were made as the references (Figures R17c and R17d). Notably, to eliminate the potential effect of $\text{WSe}_{1.8}\text{Te}_{0.2}$ nanosheet thickness on the performance, we specially selected about 10-nm-thickness nanosheets as the target catalysts (Figure R18).

As compared with MoS_2 catalysts, similar phenomena were also observed in $\text{WSe}_{1.8}\text{Te}_{0.2}$ catalysts. Figures R17e and R17f show those microcells' polarization curves and corresponding Tafel slopes, respectively. First, the vertical microcells perform similarly to in-plane microcells with full-open windows, which are superior to the in-plane half-open microcells. Second, in the vertical microcells, the ones with graphene substrates have enhanced OER performance compared to Au substrates due to their low contact barrier. Finally, tens of microcells further confirm the remarkable reproducibility of the measurements in vertical ones (Figure R17g), demonstrating that such a vertical strategy is also suitable for the non-metallic catalysts in OER.

Figure R14. The Raman spectroscopy of WSe_2 and $\text{WSe}_{1.8}\text{Te}_{0.2}$ nanosheets.

Figure R15. Transfer characteristics of $\text{WSe}_{1.8}\text{Te}_{0.2}$ nanosheet. **a**, The optical image of the field effect transistor device. Scale bar: $10\ \mu\text{m}$. **b**, Back-gate measurement of the $\text{WSe}_{1.8}\text{Te}_{0.2}$ nanosheet, showing strong *p*-type semiconducting characteristics. $V_{ds} = 100\ \text{mV}$. **c**, The leakage current of the device during the back-gate measurement.

Figure R16. *In-situ* electronic/electrochemical measurement of $\text{WSe}_{1.8}\text{Te}_{0.2}$ nanosheet for the oxygen evolution reaction. **a**, The measurement on the non-metallic $\text{WSe}_{1.8}\text{Te}_{0.2}$ microcells with full-open and half-open windows. For full-open (left panel) and half-open (right panel) windows, the polarization curves (black line) and corresponding *in-situ* conductance current curve (red line) are simultaneously collected. The optical

images of full-open (top) and half-open (bottom) windows are shown in the middle panel. **b**, The corresponding Tafel slopes of $\text{WSe}_{1.8}\text{Te}_{0.2}$ nanosheet microcells with full-open and half-open windows. Scale bar: 10 μm . The electrolyte is 0.5 M H_2SO_4 , and the window size is $\sim 50 \mu\text{m}^2$ for devices. $V_{\text{ds}} = 100 \text{ mV}$.

Figure R17. Vertical charge transport for $\text{WSe}_{1.8}\text{Te}_{0.2}$ nanosheet in oxygen evolution reaction. **a-d**, Optical images of various $\text{WSe}_{1.8}\text{Te}_{0.2}$ cells, including vertical cells with Au (**a**) and graphene (**b**) substrate layers, as well as in-plane cells with half-open (**c**) and full-open (**d**) windows. All scale bars are 10 μm . **e-f**, Polarization curves of the current density (**e**) and the corresponding Tafel slopes (**f**) for those four kinds of $\text{WSe}_{1.8}\text{Te}_{0.2}$ cells. **g**, Statistical measurement data of onset potentials and Tafel slopes obtained from tens of microcells for the OER process.

Figure R18. AFM images of $\text{WSe}_{1.8}\text{Te}_{0.2}$ nanosheets in microcells. The thickness of the target catalyst is kept as close as possible to about 10 nm.

Reviewer #3:

I have thoroughly read the manuscript entitled “The practice of reaction window in an electrocatalytic on-chip microcell”. The manuscript shows that the authors compared the effects of full-open and half-open windows on the electronic and electrochemical performance of MoS₂ nanosheets. Then, a graphene-based vertical strategy is adopted to eliminate metal-semiconductor contact, which shortens the charge pathway and achieves remarkable reproducibility in measurements. However, there remain some problems in this manuscript. Thus, this manuscript is recommended to be published in Nature Communications after major revision. Please check the details in the following:

Response: We highly appreciate the reviewer’s positive comments on our work.

Comment 1. There may be a clerical error in the legend of Figure 1c. And there are some other grammatical and format mistakes in this manuscript. The authors should check it carefully.

Response: We thank the reviewer for the careful suggestion. We have corrected the error in the legend of Figure 1c. Following the suggestion, we have also gone through the whole manuscript and corrected the grammatical and format mistakes in our revised manuscript.

Comment 2. The legend of Supplementary Figure 2a may be inaccurate. The tungsten needle is just like a wire connected to the working electrode.

Response: Thanks for the suggestion. We have revised and marked it in yellow, as shown below:

In Figure 2a in Supplementary Information: “One tungsten needle acted as the wire connected to the working electrode (WE), and the other was used to apply the drain-source voltage (V_{ds}).”

Comment 3. “TMD” in the manuscript is not assigned a full name. Please provide the full name of TMD.

Response: Thanks for the careful suggestion. We have supplemented the full name of the TMD (*i.e.*, transition metal dichalcogenides) when it first appeared.

Comment 4. The authors mentioned that “Importantly, it can achieve a high conductance, for example, 2.76 Ω mm at -0.10 V (vs. RHE) and 0.60 Ω mm at -0.55 V (vs. RHE) ...”. How are these resistor values calculated? It is better to write out the calculation process.

Response: Thanks for the suggestions. In our work, the resistivity calculation formula is given by²³,

$$\rho = \frac{R_{CH}S}{L} = \frac{V_{ds}Wt}{IL} \quad (1)$$

where ρ is the resistivity of the nanosheet, R_{CH} is the channel resistance, S is the cross-sectional area ($= W \times t$, where W is the width and t is the thickness of the nanosheet), V_{ds} is the drain-source voltage, I is the measured current value, and L is the channel length.

Taking the monolayer MoS₂ full-open microcell as the example in Figure 2a, when the electrochemical potential is -0.55 V (vs. RHE), it can be seen that: $V_{ds} = 50$ mV, $I = 0.15$ μ A, $W = (W_1 + W_2)/2 = 18$ μ m, $t = 1$ nm, $L = 10$ μ m. Finally, ρ is calculated to be 0.60 Ω mm.

We add those calculation processes in Note 1 in Supplementary Information.

Comment 5. It is suggested to add the experiment of the full-open window and half-open window on Pt metal film as a comparison.

Response: Thanks for the constructive suggestions. We made the Pt film on-chip microcells and performed the *in-situ* electronic/electrochemical measurements in both full-open and half-open microcells, as shown in Figure R19. Those results indicate that EDL regulates neither Pt film’s conductivity nor electrocatalytic activity during the reaction process due to its metallic properties.

Figure R19. *In-situ* electronic/electrochemical measurement of Pt film. **a**, The measurement on the Pt film microcells with full-open and half-open windows. For full-open (left panel) and half-open (right panel) windows, the polarization curves (black line) and corresponding *in-situ* conductance current curve (red line) are simultaneously collected. The optical images of full-open (top) and half-open (bottom) windows are shown in the middle panel. **b**, The corresponding Tafel slopes of Pt film microcells with full-open and half-open windows. Scale bar: 10 μm. The electrolyte is 0.5 M H₂SO₄, and the window size is ~160 μm² for devices. $V_{ds} = 1$ mV.

Comment 6. More related papers on single nanosheet electrochemical devices should be noticed. It is suggested that the authors cite the following relevant articles to make the manuscript more convincing.

Peiyao Wang et.al. Oxygen evolution reaction dynamics monitored by an individual nanosheet-based electronic circuit, *Nature Communications*, 2017, 8, 645.

Jiazhao Huang et.al. Back-Gated van der Waals Heterojunction Manipulates Local Charges toward Fine-Tuning Hydrogen Evolution. *Angew. Chem. Int. Ed.*, 2022, 61, e202203522.

Mengyu Yan et.al. Field-Effect Tuned Adsorption Dynamics of VSe₂ Nanosheets for Enhanced Hydrogen Evolution Reaction, *Nano Letters*, 2017, 17, 4109–4115.

Response: Thanks for the reviewer’s suggestion, which makes our manuscript more convincing in the on-chip microcell field. We have cited the relevant articles in the revised manuscript.

References:

1. Ferain, I., Colinge, C.A. & Colinge, J.-P. Multigate transistors as the future of classical metal–oxide–semiconductor field-effect transistors. *Nature* **479**, 310-316 (2011).
2. He, Y., *et al.* Self-gating in semiconductor electrocatalysis. *Nat. Mater.* **18**, 1098-1104 (2019).
3. He, Y., *et al.* Amorphizing noble metal chalcogenide catalysts at the single-layer limit towards hydrogen production. *Nat. Catal.* **5**, 212-221 (2022).
4. Liu, G., *et al.* Hydrogen evolution reaction on in-plane platinum and palladium dichalcogenides via single-atom doping. *Int. J. Hydrogen Energy* **46**, 18294-18304 (2021).
5. Lee, J., *et al.* Hydrogen Evolution Reaction at Anion Vacancy of Two-Dimensional Transition-Metal Dichalcogenides: Ab Initio Computational Screening. *J. Phys. Chem. Lett.* **9**, 2049-2055 (2018).
6. Seh, Z.W., *et al.* Combining theory and experiment in electrocatalysis: Insights into materials design. *Science* **355**, eaad4998 (2017).
7. Liu, Y., *et al.* Approaching the Schottky–Mott limit in van der Waals metal–semiconductor junctions. *Nature* **557**, 696-700 (2018).
8. Shen, P.-C., *et al.* Ultralow contact resistance between semimetal and monolayer semiconductors. *Nature* **593**, 211-217 (2021).
9. Thakurta, S.R.G. & Dutta, A.K. Electrical conductivity, thermoelectric power and hall effect in p-type molybdenite (MoS₂) crystal. *J. Phys. Chem. Solids* **44**, 407-416 (1983).
10. Xia, H., Shi, Z., Gong, C. & He, Y. Recent strategies for activating the basal planes of transition metal dichalcogenides towards hydrogen production. *J. Mater. Chem. A* **10**, 19067-19089 (2022).
11. Li, H., *et al.* Activating and optimizing MoS₂ basal planes for hydrogen evolution through the formation of strained sulphur vacancies. *Nat. Mater.* **15**, 48-53 (2016).
12. Lai, B., *et al.* Hydrogen evolution reaction from bare and surface-functionalized few-layered MoS₂ nanosheets in acidic and alkaline electrolytes. *Mater. Today Chem.* **14**, 100207 (2019).
13. Ye, G., *et al.* Defects Engineered Monolayer MoS₂ for Improved Hydrogen Evolution Reaction. *Nano Lett.* **16**, 1097-1103 (2016).
14. Luo, Y., *et al.* Two-Dimensional MoS₂ Confined Co(OH)₂ Electrocatalysts for Hydrogen Evolution in Alkaline Electrolytes. *ACS Nano* **12**, 4565-4573 (2018).
15. Wu, W., *et al.* Activation of MoS₂ Basal Planes for Hydrogen Evolution by Zinc. *Angew. Chem., Int. Ed.* **58**, 2029-2033 (2019).
16. He, M., *et al.* Enhanced hydrogen evolution reaction activity of hydrogen-annealed vertical MoS₂ nanosheets. *RSC Adv.* **8**, 14369-14376 (2018).
17. Ren, X., Yang, F., Chen, R., Ren, P. & Wang, Y. Improvement of HER activity for MoS₂: insight into the effect and mechanism of phosphorus post-doping. *New J. Chem.* **44**, 1493-1499 (2020).
18. Xia, K., Cong, M., Xu, F., Ding, X. & Zhang, X. Targeted Assembly of Ultrathin NiO/MoS₂ Electrodes for Electrocatalytic Hydrogen Evolution in Alkaline Electrolyte. *Nanomaterials*, **10**, 1547 (2020).
19. Geng, X., *et al.* Pure and stable metallic phase molybdenum disulfide nanosheets for hydrogen evolution reaction. *Nat. Commun.* **7**, 10672 (2016).
20. Luo, Z., *et al.* Chemically activating MoS₂ via spontaneous atomic palladium interfacial doping towards efficient hydrogen evolution. *Nat. Commun.* **9**, 2120 (2018).
21. Zhu, J., *et al.* Boundary activated hydrogen evolution reaction on monolayer MoS₂. *Nat. Commun.* **10**, 1348 (2019).
22. Yang, J., *et al.* Ultrahigh-current-density niobium disulfide catalysts for hydrogen evolution. *Nat. Mater.* **18**, 1309-1314 (2019).

23. Mitta, S.B., *et al.* Electrical characterization of 2D materials-based field-effect transistors. *2D Mater.* **8**, 012002 (2021).

REVIEWERS' COMMENTS

Reviewer #1 (Remarks to the Author):

The authors have addressed my concern well. After careful consideration, I believe this work is now suitable to publish in Nat. Commun.

Reviewer #2 (Remarks to the Author):

This manuscript can be accepted.

Reviewer #3 (Remarks to the Author):

The authors have addressed the comments/suggestions carefully. I am satisfied with the revisions. This manuscript is now acceptable for publication.